# Complex interactions in healthcare expenditure through the years: A panel data analysis using fsQCA in OECD countries with policy implications

**Shanren Nie[1], Dong Liu[2], Sheng Chen**[1]*

**1** School of Public Policy and Administration, Chongqing University, Chongqing, Chongqing, China,
**2** Department of Social Work at Chongqing Intellectual Property School, Chongqing University of Technology, Chongqing, Chongqing, China

* chensh8623@cqu.edu.cn

**Funding:** Project Nos. 2020CDJSK01WT07, 2023CDJSKPT04 supported by the Fundamental Research Funds for the Central Universities,

## Abstract

This paper presents a novel longitudinal study examining interactions in healthcare expenditure (HCE). Utilizing fuzzy-set qualitative comparative analysis (fsQCA), the study constructs a consumption-provision-finance model to investigate factors influencing HCE. Data from the Organization for Economic Cooperation and Development (OECD) database for the years 2010-2022 are analyzed, covering 37 OECD countries (excluding Luxembourg due to insufficient data). By treating each country-year OECD observation as a case, causal recipes are identified and interpreted. The characteristics of the panel data set are examined by assessing the stability of causal patterns over multiple years and exploring the consistency of these patterns within individual countries across various years. The findings of this study offer significant insights for policy management and future research, particularly in relation to the diverse annual relationships observed between factors and HCE in different countries. Additionally, comparisons are drawn between panel fsQCA and cross-sectional fsQCA conducted for specific years.

## Background

The healthcare system of a country is crucial not only for the well-being of its citizens but also for its economic prosperity [1], since effective healthcare provision enhances human capital and boosts productivity [2]. However, there is a significant challenge to keeping healthcare systems running safely and effectively. Some countries face high healthcare expenditure (HCE) [3], while others struggle with rapid HCE growth, raising concerns about their ability to handle future challenges, such as an aging population.

Given these diverse scenarios, it is both practically and theoretically beneficial to answer the question: *What factors influence the HCE most through the year?* Understanding what influences HCE can inform strategies for managing healthcare costs while ensuring the system remains robust and responsive. A comparative approach, examining HCE differences geographically and historically, can help identify these factors. By comparing HCE across

National Natural Science Foundation of China
No. 72274026.

**Competing interests:** The authors have declared that no competing interests exist.

different countries and over different time periods within a single country, we can uncover the underlying factors that explain these differences.

Fig 1 illustrates the trend of HCE (as a percentage of GDP) among thirty-seven countries from 2010 to 2022. Geographically, there is a notable disparity in HCE levels; countries like the United States consistently exhibit high HCE, above 16% of GDP, while others, such as Turkey, maintain a significantly lower HCE, below 5% of GDP. Historically, while most nations have not experienced substantial growth in HCE during these years, a clear fluctuation in HCE is observed across all countries starting from 2020, likely due to the impact of the COVID-19 pandemic. An intriguing observation is the variability in HCE trends post-2019. For some countries, such as Austria and Czechia, HCE levels have remained stable. In contrast, countries like Chile and South Korea have shown a continuous upward trend, while others, including Ireland and Costa Rica, have experienced a decline in HCE. These comparative findings may yield valuable insights into the factors influencing HCE.

The OECD provides a unified framework for comparing international health spending, encompassing both total expenditure and the structure of spending on healthcare goods and services [4]. As shown in Fig 2, this framework includes three dimensions: consumption, provision, and financing of healthcare. This categorization offers a comprehensive view of healthcare expenditure, aiding in understanding the dynamics of different healthcare systems.

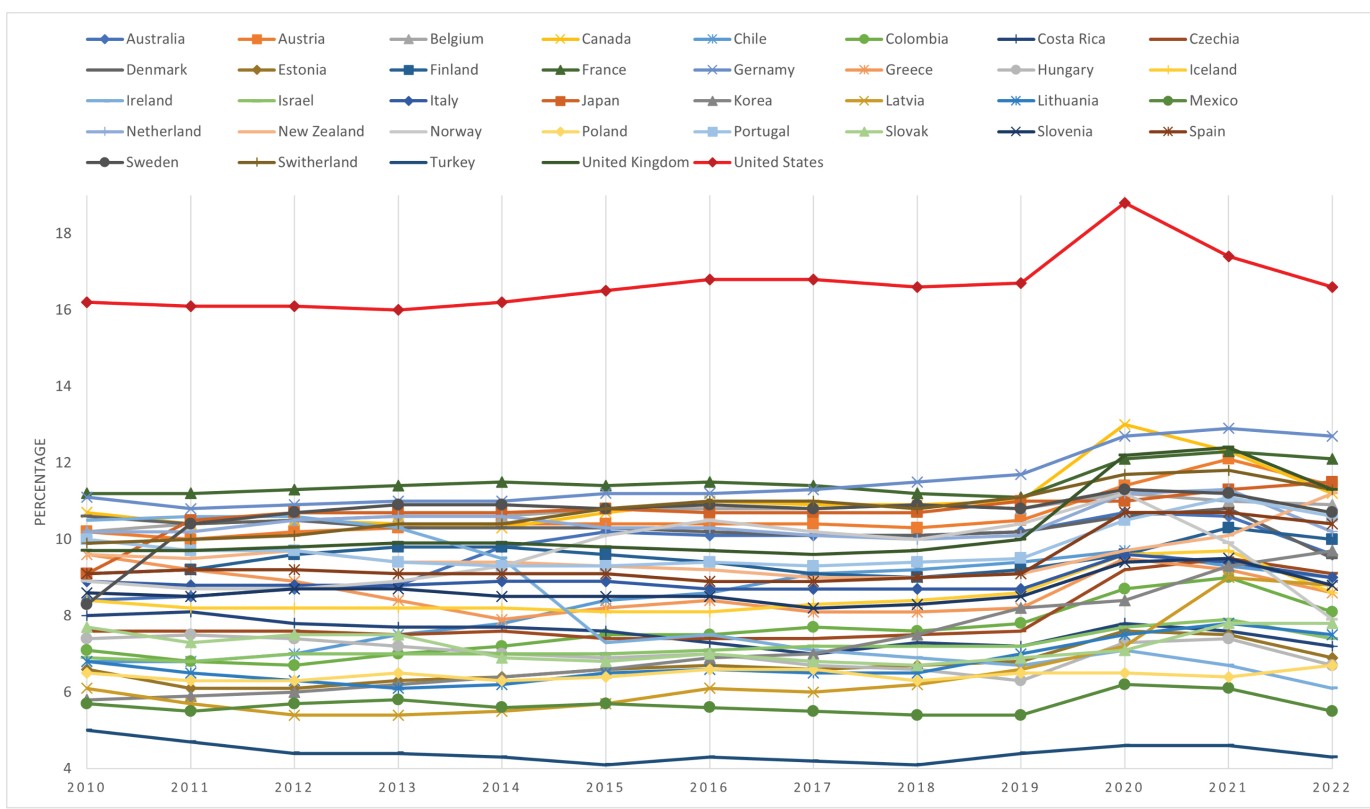

**Fig 1. Healthcare Expenditure as percentage of GDP in OECD, 2010–2022.** This figure shows healthcare expenditure as a percentage of GDP across OECD countries from 2010 to 2022.

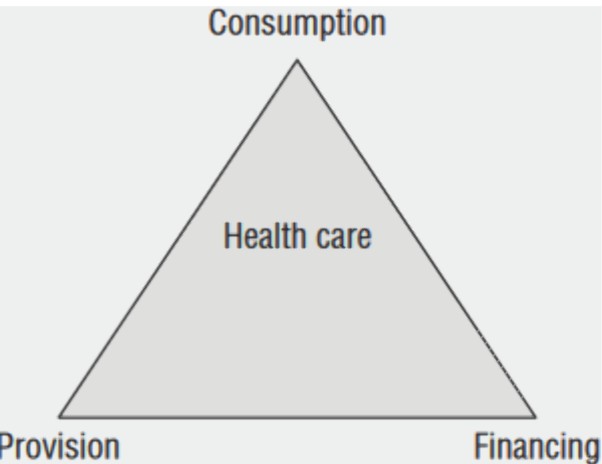

**Fig 2. Healthcare accounts framework.** This figure presents the Healthcare accounts framework, which is composed of three key dimensions: consumption, provision, and financing.

Given the complexity of factors influencing HCE, it is reasonable to explore the combined effects of multiple factors. This study conducts a panel data analysis using fuzzy set Qualitative Comparative Analysis (fsQCA) to identify the configurations of factors influencing HCE over time.

## Factors of healthcare expenditure: A brief review of the literature

Extensive literature exists on HCE. The data used in these studies can be either microeconomic or macroeconomic [5]. It's crucial to distinguish between them, as Parkin criticized Newhouse's use of macro data for microeconomic concepts [6]. This study uses macroeconomic data.

Reviews of HCE determinants survey various models and findings. Based on these studies, here is a list of commonly recognized important factors.

**Income.** There is a well-established link between national HCE and income [6]. Newhouse confirmed that per capita medical care expenditure is strongly related to per capita income across countries [7]. Hitiris reaffirmed GDP's importance in health spending using a sample of 560 pooled time-series and cross-section observations [8].

Advancements in research methods have allowed for more precise analyses. Okunade used the Box-Cox model to study 26 African countries, finding that economic determinants accounted for 74% of health expenditure variations [9]. Di Matteo's research on American state and Canadian province data showed income explains a small portion of health expenditures when controlling for time effects [10]. Martin's review found only four out of 20 articles considered income the principal HCE determinant [11].

Panel data has strengthened the observed impact of income on HCE. Ke [5] expanded the range of factors contributing to HCE variation using data from 143 countries. Furuoka [12] also identified a significant relationship between GDP and HCE in twelve Asian countries. Samadi [13] discovered the short and long term correlation between GDP per capita and HCE by using panel data econometric methods. Hartwig confirmed GDP growth as a significant HCE growth determinant in 33 OECD countries [14]. Bedir, Akca, Barkat, Baradi, and Azwardi provided further evidence of income's influence on HCE across various regions and income levels [15–19].

**Aging population/Demographic structure.** Population age structure is often included as a covariate in health expenditure regressions, with indicators like the share of people under 15 and over 65 [5]. Mendelson [20] confirmed that both aging and population growth significantly contribute to the rise in HCE by investigating the relative costs of treating patients aged 65 and over. However, Barros [21] studied 24 OECD countries and found that health system characteristics such as population aging did not significantly influence HCE growth. Tchoe [22] further confirmed that population aging is not a significant determinant of HCE based on an econometric analysis using OECD health data and time-series data for Korea, noting that the contribution of population aging to HCE is less than 10%.

Later literature reconsidered the impact of an aging population on HCE. Studies have shown that per-capita spending on healthcare is substantially higher for individuals aged 65 and older compared to younger individuals, with the disparity increasing consistently as the average age exceeds 65 years [23]. Li [24] confirmed that the age effect on HCE for the population aged 65 or over was the most significant among other age groups. Similarly, Yang [25] stated that population aging leads to increased HCE.

**Lifestyle-related factors.** Only a few studies conducted in OECD countries and the U.S. have explored the association between HCE growth and lifestyle-related factors such as alcohol and cigarette consumption, calorie intake, sugar consumption, obesity, and physical activity [26]. The scarcity of data might explain the limited research in this area. Despite this, Thornton's study provided insights into how less healthy lifestyles could contribute to high levels of HCE. By analyzing data from all 50 U.S. states for the year 1998, Thornton indicated that unhealthy lifestyles are a significant factor in HCE [27]. Furthermore, Thornton [28] identified excessive alcohol consumption as one of the most important factors influencing HCE by employing a panel data model for the period 1993-2009. In conclusion, less healthy lifestyles have been identified as significant factors contributing to high levels of HCE.

**Technological progress.** Technological progress has been recognized as a significant driver of healthcare expenditure. Martin [11] reviewed the literature on the determinants of HCE from 1998 to 2007 and found that factors such as technological progress are increasingly considered important in explanatory models of HCE. You [29] further emphasized the global impact of healthcare technology by estimating its effects on HCE. Several studies have identified technological progress as a major determinant of HCE [30–33]. Marino [34] presented a systematic review of the impact of technological advancements on health spending, introducing various approaches to measuring the contribution of technological change to HCE growth. Collectively, the literature supports the conclusion that technological progress is a significant driver of growth in health expenditure, with estimates of its relative contribution to HCE growth ranging from 10% to 75%.

**Healthcare price.** According to Hitiris [35], the rise in healthcare expenditure (HCE) results from consumer and producer choices underlying demand and supply. His argument is based on the premise that an individual's demand for healthcare depends on their health status, income, the price of healthcare, and health insurance. Gerdtham [36] found that healthcare prices had the greatest impact on HCE in countries with mixed healthcare systems by studying 25 countries with different types of healthcare systems. However, Bac [37] argued that the price of healthcare does not significantly affect the amount of resources devoted to healthcare. Wang [38] confirmed a cointegration relationship between HCE and the relative price of healthcare for most countries. Conversely, Raeissi [39] stated that HCE is not necessarily higher in countries with higher healthcare prices.

**Healthcare system characteristics.**

- Healthcare provision. Many factors in healthcare provision have been studied to determine their relation to HCE, such as the number of hospital beds [11,26,38,40], number of doctors/nurses [41], and number of health institutions [42]. These factors' influence on HCE are also controversial; some studies found that more provision of healthcare products and services seems to result in higher HCE [13,43], while others found the opposite [40,44].
- Provider payment mechanisms. Fee-for-service systems tended to lead to higher HCE on average than other payment system such as capitation systems [40]. This has lead to payments mechanism reform in many countries.

Based on the literature review, we can draw the following conclusions: Common factors influencing the level and growth of total healthcare expenditures include income (per capita GDP), aging population dynamics, technological progress, healthy lifestyle, and variations in healthcare system characteristics. Recent studies also underscore the critical importance of understanding both the demand and supply of healthcare products and services.

## Methods

### Model presentation and factors selection

According to the OECD [4], healthcare expenditure can be categorized into three axes: consumption, provision, and financing (see Fig 2). Based on this, we developed a consumption-provision-finance model to select factors within these dimensions. The factors are income, aging population, unhealthy population (lifestyle-related factors), healthcare provision, and healthcare price. Each factor and its corresponding dimension is shown in Table 1. We focused on direct influence factors, excluding technological progress, as its impact is captured through other factors like healthcare price.

### Method, data, and calibration

**Method.** In this study, fuzzy-set QCA (fsQCA) is employed to identify configurations of factors that influence HCE [45]. fsQCA is a configurational method that combines qualitative and quantitative approaches to identify causal patterns leading to specific outcomes. Unlike traditional regression-based methods, fsQCA focuses on set relations and accounts for causal complexity, allowing for multiple pathways to the same result. The process of fsQCA involves several key steps, including calibration of data into fuzzy sets, necessity analysis to determine whether a condition is required for the outcome, and sufficiency analysis to identify combinations of conditions that produce the outcome. By using fuzzy sets, fsQCA captures varying degrees of membership in causal conditions, making it particularly useful for examining complex configurations of factors that jointly contribute to an outcome.

**Table 1. The conditions in three dimensions of healthcare accounting framework.**

| Consumption conditions | Provision conditions | Financing conditions |
| --- | --- | --- |
| Aging population | Healthcare provision | Income |
| Unhealthy population | Healthcare prices | |

The healthcare account framework consists of three dimensions: consumption, provision, and financing. For consumption, the conditions are an aging population and an unhealthy population. In the provision dimension, healthcare provision levels and prices are considered. Income is the condition for the financing dimension.

After examining the interactions between factors and HCE, Garcia-Castro's method explores these interactions over time [46]. This method focuses on consistency-oriented developments, specifically pooled consistency (POCONS), between consistency (BECONS), and within consistency (WICONS). POCONS evaluates overall consistency by aggregating all time points, BECONS examines stability across different time periods, and WICONS assesses consistency within each time point separately. These measures provide a comprehensive understanding of the temporal dynamics in set relations, ensuring that causal patterns remain robust over time. These measures provide a promising approach to incorporating time into fsQCA, facilitating longitudinal set-theoretic research.

**Data.** The data were primarily sourced from the OECD database, covering 37 OECD countries from 2010 to 2022, resulting in a total of 481 country-year observations. Luxembourg was excluded due to insufficient data availability. Unhealthy population data were obtained from the World Health Organization (WHO) database. Missing values in isolated years were linearly interpolated using R's approx() (rule=1 for no extrapolation), following best practices for small temporal gaps (<10%). Sensitivity analyses confirmed robustness [47,48].

**Calibration.** The first step of fsQCA is the calibration of the conditions and outcome. This section reports the calibration process. The calibration undertaken here follows the direct method introduced by Dusa [49]. For each indicator of the five conditions and one outcome, three thresholds need to be determined: the complete exclusion point, the crossover point, and the complete inclusion point, which are obtained using the findTh() function in the QCA package. The only exception is income (GDP per capita in USD). The thresholds for income are derived from the World Bank's definition of high-income country categories in 2023. Specifically, the middle-high-income level is used as the complete exclusion point, the high-income level as the crossover point, and a computed value equidistant from the crossover point to the complete exclusion point for the complete inclusion point, thereby establishing the three thresholds for income. Subsequently, the logistic distribution function, known as the "s-shape" function in QCA, is used to set membership values. The threshold values for all indicators' calibration are presented in Table 2.

## fsQCA analysis

The initial fsQCA analysis follows the standard iterative approach, encompassing both necessity and sufficiency analyses. The sufficiency analysis incorporates the use of a truth table. Given the asymmetric nature of fsQCA, it is essential to analyze High-HCE and Low-HCE separately. The following sections outline the steps in detail.

**Table 2. Thresholds of outcome and conditions for calibration.**

| Set | Indicators | Exclusion | Crossover | Inclusion |
|---|---|---|---|---|
| HCE | HCE as a share of GDP (HCE) | 6.4 | 10.3 | 14.5 |
| Income | GDP per capita in USD (GDP) | 10462 | 48220 | 85978 |
| Aging population | Population at age 65 or over (OLD) | 10.5 | 17.3 | 24.6 |
| Unhealthy population | Population of obese (UH) | 8.5 | 18 | 29.1 |
| Healthcare provision | Active doctors per 1000 inhabitants (DOC) | 2 | 4.2 | 5.2 |
| Healthcare price | Price level in healthcare sector (P) | 41.5 | 82 | 113 |

The exclusion, crossover, and inclusion thresholds for calibrating the outcome and the five conditions.

## Necessity analysis results

In fsQCA, the analysis of necessary conditions is conducted independently to determine if a single condition is essential for the outcome. Using the standard threshold of 0.9, the results in Table 3 reveal that no single condition is necessary for either High-HCE or Low-HCE.

## Sufficiency analysis results

The sufficiency analysis focuses on understanding configurations of conditions in relation to High-HCE or Low-HCE. This process aims to identify configurations of conditions that are sufficient for the outcome to occur. Central to this approach is the use of a truth table, which enumerates all possible configurations of conditions (five in this case), assigning them binary values (0 for absence, 1 for presence). For each country-year observation, conditions are assigned a membership score based on their strength of association (<= or >= 0.5).

Each configuration is characterized by several metrics, including the number of country-year observations strongly associated with it, and for High-HCE and Low-HCE, the consistency level. To determine which configurations merit further consideration due to their assured association with either High-HCE or Low-HCE, two thresholds must be met: frequency and consistency.

The frequency threshold requires at least 8 country-year observations for a configuration to be considered further, applicable to both High-HCE and Low-HCE. For High-HCE, a consistency threshold of 0.7 is set, while for Low-HCE, it is 0.977. These thresholds are chosen to ensure distinct associations between configurations and outcomes.

As a result of these threshold choices, nine configurations (covering 235 country-year observations) are associated with High-HCE, and four configurations (covering 199 country-year observations) are linked to Low-HCE, totaling 434 out of 481 country-year observations.

The remaining configurations are termed remainders, lacking sufficient empirical evidence to support their association with specific outcomes. Although they do not show a clear association with High-HCE or Low-HCE, they contribute to understanding different configurations linked to specific outcomes.

The immediate analysis presented here employs a directional expectation, suggesting that high income, a significant proportion of aging population, a prevalence of unhealthy population, extensive healthcare provision, and high healthcare prices contribute to High-HCE. Conversely, low income, a smaller proportion of aging population, a lower prevalence

**Table 3. Analysis of necessity for High-HCE and Low-HCE.**

| Condition | | High-HCE | | Low-HCE | |
|---|---|---|---|---|---|
| | | Consistency | Coverage | Consistency | Coverage |
| GDP | High | 0.846 | 0.706 | 0.468 | 0.809 |
| | Low | 0.771 | 0.412 | 0.830 | 0.918 |
| OLD | High | 0.843 | 0.577 | 0.557 | 0.790 |
| | Low | 0.693 | 0.431 | 0.702 | 0.902 |
| UH | High | 0.745 | 0.386 | 0.787 | 0.843 |
| | Low | 0.697 | 0.613 | 0.426 | 0.776 |
| DOC | High | 0.625 | 0.608 | 0.419 | 0.844 |
| | Low | 0.840 | 0.411 | 0.805 | 0.816 |
| P | High | 0.830 | 0.554 | 0.490 | 0.676 |
| | Low | 0.514 | 0.327 | 0.677 | 0.892 |

The consistency and coverage of the five conditions for both High-HCE and Low-HCE outcomes. Since the consistency of none of the conditions exceeds 0.9, it indicates that no single condition is necessary for either outcome.

of unhealthy population, limited healthcare provision, and low healthcare prices result in Low-HCE. The immediate solution reveal causal recipes: configurations of conditions to particular outcomes. These findings are denoted using circle notation adapted from Ragin ("○" for absence and "●" for presence of a condition) [45].

Table 4 summarizes the results of the sufficiency analysis, identifying three (MH1-MH3) causal recipes for High-HCE and two (ML1 and ML2) for Low-HCE. Further analysis is needed to explore the countries covered by these recipes and the stability of these associations over time.

## Panel data breakdown of fsQCA results

This section builds on the fsQCA results by exploring findings across different years and countries using panel data. Three consistency measures are employed:

1. POCONS: Covers all observations, matching the previously presented consistency values (see Table 4).
2. BECONS: Focuses on specific year observations.
3. WICONS: Pertains to specific country observations.

Analyses with BECONS and WICONS are detailed separately for the causal recipes linked to High-HCE and Low-HCE outcomes.

### High-HCE

Table 4 presents three causal recipes (MH1, MH2, and MH3) associated with High-HCE. Their corresponding BECONS graphs are depicted in Fig 3. For each causal recipe, the BECONS results for MH1, MH2, and MH3 are shown across individual years included in the dataset, with the corresponding POCONS values indicated as a horizontal dashed line.

The three BECONS graphs exhibit similarities, reflecting the shared characteristics of the causal recipes. For example, as shown in Table 4, configuration 28 is associated with both MH1 and MH3 causal recipes. There is notable variability in BECONS values in more

**Table 4. Sufficiency analysis of GDP, OLD, UH, DOC and P with High-HCE and Low-HCE outcomes.**

| Conditions | HCE | | | | |
| --- | --- | --- | --- | --- | --- |
| | High | | | Low | |
| GDP | | | ● | ○ | ○ |
| OLD | ● | | ● | | ○ |
| UH | ○ | ● | | ● | |
| DOC | | | ● | | ○ |
| P | | ● | | ○ | ○ |
| Immediate Solution | MH1 | MH2 | MH3 | ML1 | ML2 |
| Configurations(In strong membership terms) | 28,26,10,9 | 30,14,22,6 | 28,31 | 5,13,15 | 5,1 |
| Consistency* | 0.750 | 0.637 | 0.839 | 0.975 | 0.979 |
| Raw coverage* | 0.663 | 0.649 | 0.582 | 0.528 | 0.411 |
| Unique coverage* | 0.119 | 0.199 | 0.019 | 0.164 | 0.047 |
| Solution consistency* | 0.612 | | | 0.969 | |
| Solution coverage* | 0.937 | | | 0.576 | |

* The consistency and coverage values are over the whole data set of cases.
The sufficiency analysis results identify three causal recipes (MH1-MH3) for High-HCE and two (ML1 and ML2) for Low-HCE.

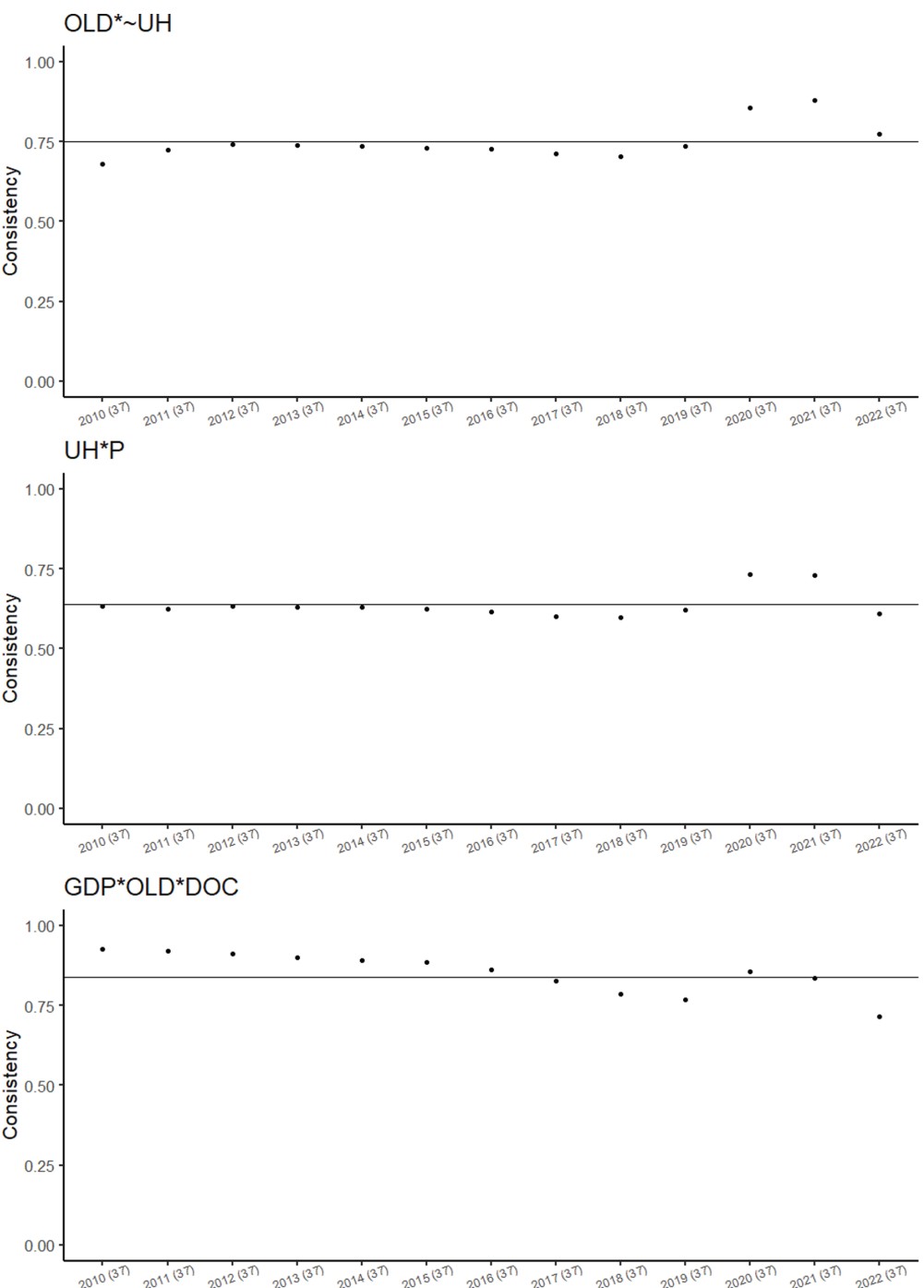

**Fig 3. BECONS values for causal recipes MH1, MH2 and MH3 (High-HCE).** In this figure, each subplot individually presents the BECONS values of causal recipes MH1, MH2, and MH3 for the High-HCE outcome throughout the years.

recent years (2019 onwards). Specific years where MH1 and MH3 BECONS values differ are before 2019 ($MH1<MH3$), and from 2020 onwards ($MH1>MH3$). As discussed later, these differences may be influenced by reactions to the COVID-19 pandemic, causing fluctuations in configurations and associated recipes before stabilizing post-pandemic.

WICONS results are analyzed at the country level, reflecting the consistency of specific causal recipes within each country. This consistency is influenced by whether a country strongly adheres to one or more configurations during the years it is included in the dataset, as elaborated further. The corresponding WICONS values for all 37 countries are illustrated in Fig 4.

In each graph in Fig 4, the horizontal axis represents the rank of countries' WICONS values, which are ordered separately for each of the three causal recipes, resulting in a distinct ranking each time. The relevant POCONS values are depicted as dashed lines. Within each graph, three WICONS values are displayed for each country to facilitate comparisons across causal recipes.

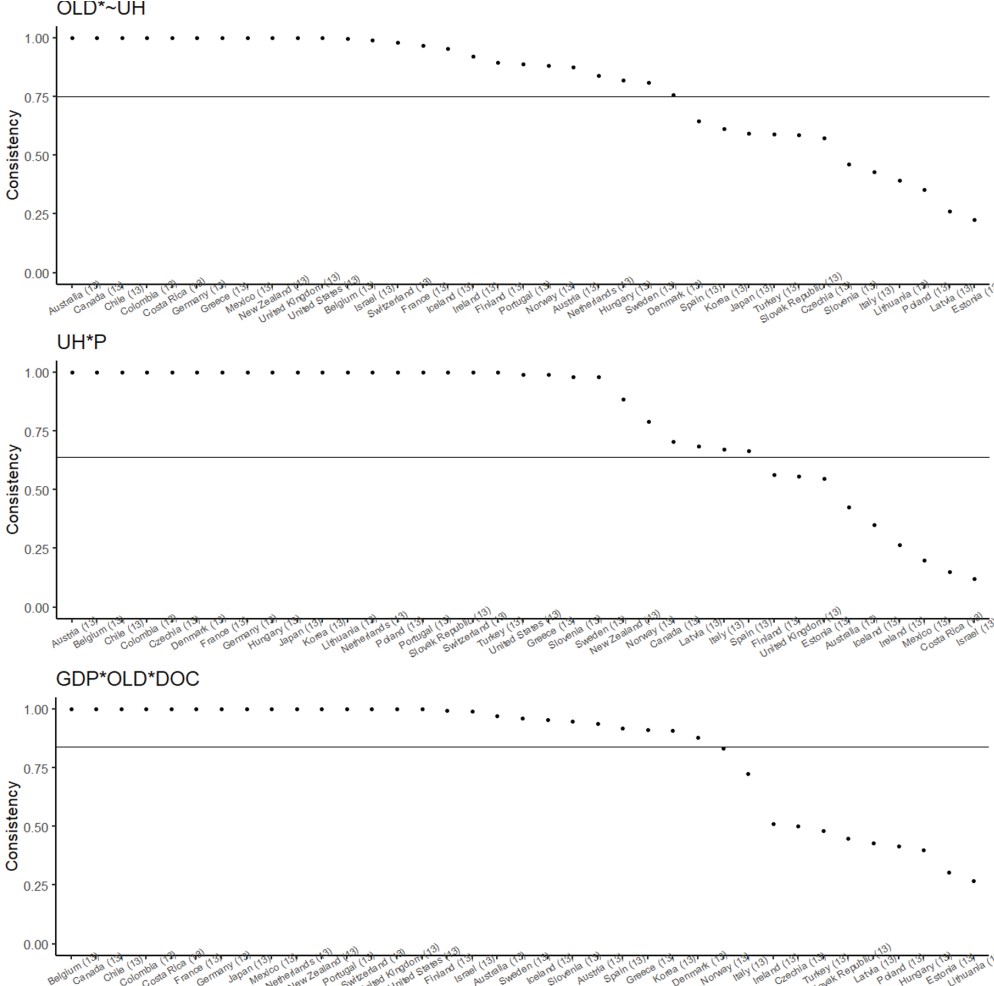

**Fig 4. WICONS values for all countries for causal recipes MH1, MH2 and MH3 (High-HCE).** In this figure, each subplot individually presents the WICONS values of causal recipes MH1, MH2, and MH3 for the High-HCE outcome for each country.

A significant observation is the initiation of consistent results (WICONS = 1.000 within rank-ordered values): approximately 12 countries exhibit this consistency for MH1, 18 countries for MH2, and 15 countries for MH3.

Of particular interest are the results on the right-hand side of the graphs, indicating countries that do not consistently adhere to a specific causal recipe across the years covered by the 481 country-year observations dataset. Broadly, this finding suggests strong consistency for approximately 18 of the included countries. Other countries exhibit varying levels of consistency over the analyzed years.

### Low-HCE

Table 4 presents two causal recipes associated with Low-HCE: ML1 and ML2. Their corresponding BECONS graphs are illustrated in Fig 5.

In Fig 5, the BECONS results for ML1 and ML2 are shown for each individual year in the dataset, with the corresponding POCONS values also included (see Table 4). The BECONS graphs display a general similarity, reflecting the overlap in configurations among the causal recipes considered, both of which include configuration 5. Notably, consistency values are consistently high and equal to or greater than POCONS before 2019 for both causal recipes. However, this trend starts to shift from 2019 onwards, potentially influenced by the COVID-19 pandemic.

The WICONS results pertain to the country level. Fig 6 presents the relevant WICONS values for the 37 countries in the dataset. Approximately 30 countries exhibit strong consistency

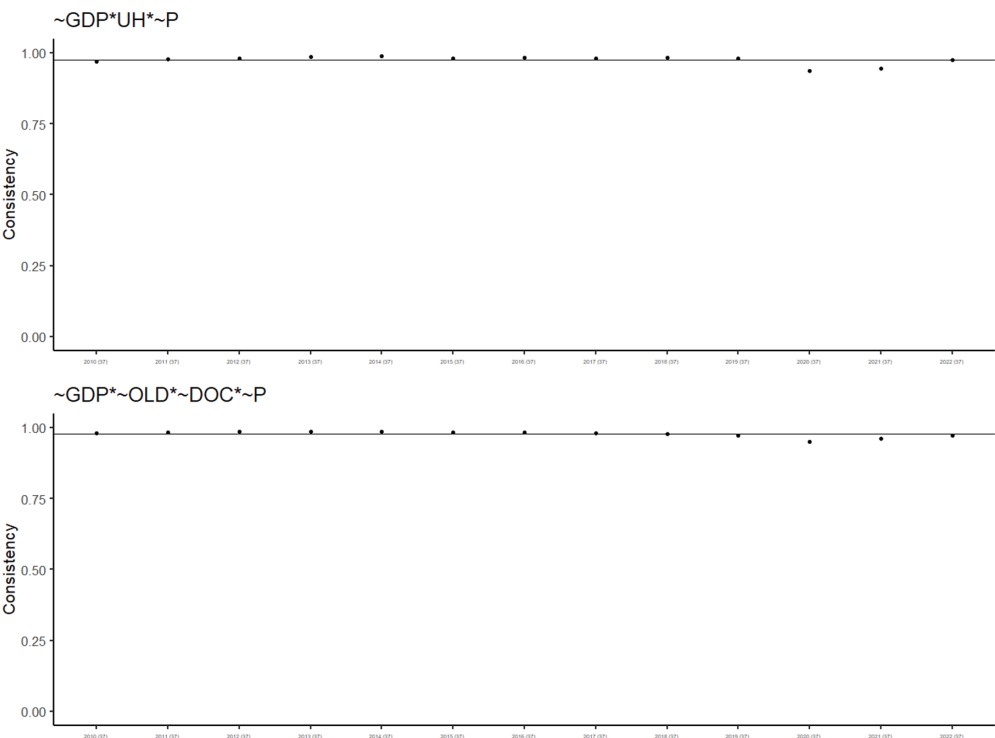

**Fig 5. BECONS values for all countries for causal recipes ML1 and ML3 (Low-HCE).** In this figure, each subplot individually presents the BECONS values of causal recipes ML1 and ML2 for the Low-HCE outcome throughout the years.

in terms of the recipes they are associated with. The remaining countries demonstrate varying degrees of inconsistency across the years included in the analysis.

Similar to the analysis for High-HCE, each of the causal recipes for Low-HCE shows consistency in 28 to 31 countries with unit WICONS consistency.

## Discussion

Building on the fsQCA findings, this section analyzes causal configurations (recipes) and examines between-country (BECONS) and within-country (WICONS) consistencies through a comparative country-level lens. The analysis reveals strong consistency in the identified causal recipes and in the BECONS and WICONS analysis of the longitudinal data.

This overall consistency suggests that the long-term impact of the COVID-19 pandemic on HCE was relatively minor for many countries, as indicated by a recovery trend in HCE. However, the changing association with recipes over time in some countries provides insight into when and how these changes occurred, relative to the configurations they are associated with. Table 4 shows the strong membership association of a sample of countries to configurations, representing all years in the dataset.

Table 5 considers a small sample of countries. For each country and each year in the dataset, the configurations they are most associated with are noted by year, along with the causal recipes linking that configuration to either High-HCE or Low-HCE. The number of configurations representing the countries over the 13 years from 2010 to 2022.

For High-HCE countries, the global COVID-19 pandemic had a significant short-term impact on HCE, but a minimal long-term effect. Austria, Belgium, Denmark, Ireland, and the United States all experienced notable increases in HCE in 2020 and beyond, followed by

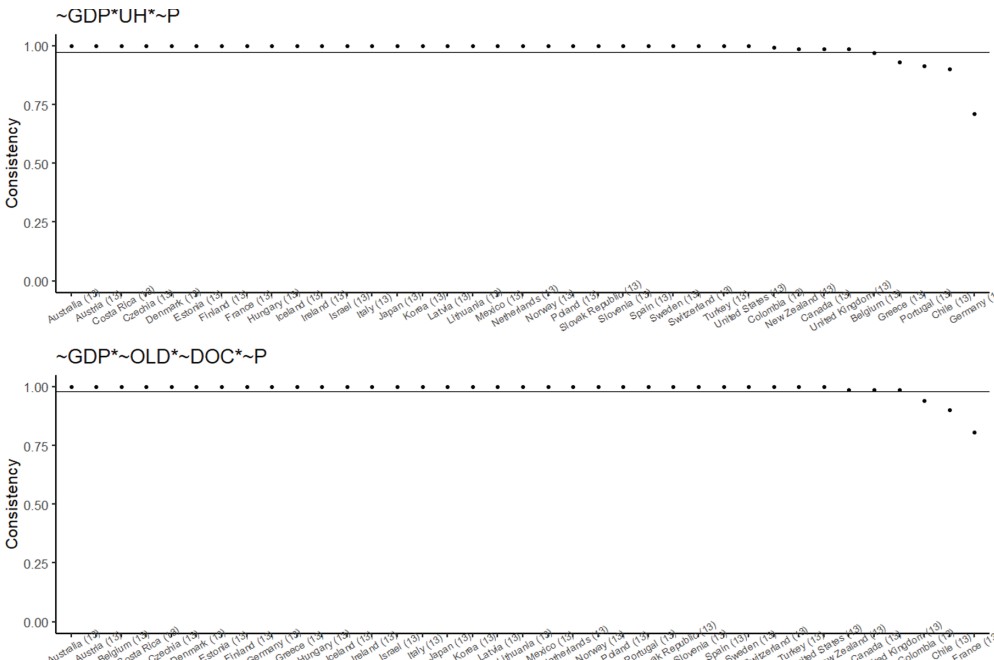

**Fig 6. WICONS values for all countries for causal recipes ML1 and ML2 (Low-HCE).** In this figure, each subplot individually presents the WICONS values of causal recipes ML1 and ML2 for the Low-HCE outcome for each country.

a decline in 2022. However, each country has a unique narrative. In Austria and Denmark, the primary High-HCE recipe, MH1, involves a large proportion of the elderly population and a low percentage of unhealthy people, resulting in a slower recovery. For Austria, Ireland, and the United States, the dominant High-HCE recipe since around 2019 has been MH2 (a high proportion of unhealthy population and high healthcare costs), leading to a faster recovery compared to the former countries.

In Austria, the High-HCE causal recipe was MH1 from 2010 to 2013, with MH3 added from 2014 to 2022, resulting in a steady rise in HCE. Although it began to decline after the

**Table 5. Breakdown of country-specific strong membership (2010–2022) [GDP, OLD, UH, DOC, P].**

| Country | Strong membership details | | | | |
|---|---|---|---|---|---|
| Austria | Configuration | [0,1,0,1,1] | [1,1,0,1,1] | | |
| | Years | 2010,2011,2012, 2013 | 2014,2015,2016, 2017,2018,2019, 2020,2021,2022 | | |
| | Causal recipe(s) | MH1 | MH1\|MH3 | | |
| Belgium | Configuration | [0,0,0,0,1] | [0,1,0,0,1] | [0,1,1,0,1] | [1,1,1,0,1] | [1,1,1,0,0] |
| | Years | 2010 | 2011,2012 | 2013,2014,2015 | 2016,2017,2018, 2019,2020,2021 | 2022 |
| | Causal recipe(s) | — | MH1 | MH2 | MH2 | — |
| Denmark | Configuration | [0,0,0,0,1] | [0,1,0,0,1] | [1,1,0,0,1] | [1,1,0,1,1] | |
| | Years | 2010,2011 | 2012,2013,2014 | 2015,2016,2017 | 2018,2019,2020, 2021,2022 | |
| | Causal recipe(s) | — | MH1 | MH1 | MH1\|MH3 | |
| Ireland | Configuration | [0,0,1,0,1] | [1,0,1,0,1] | | |
| | Years | 2010,2011,2012, 2013 | 2014,2015,2016, 2017,2018,2019, 2020,2021,2022 | | |
| | Causal recipe(s) | — | MH2 | | |
| United States | Configuration | [1,0,1,0,1] | [1,1,1,0,1] | | |
| | Years | 2010,2011,2012, 2013,2014,2015, 2016,2017,2018, 2019,2020,2021 | 2022 | | |
| | Causal recipe(s) | MH2 | MH2 | | |
| Chile | Configurations | [0,0,1,0,0] | | | |
| | Years | 2010,2011,2012, 2013,2014,2015, 2016,2017,2018, 2019,2020,2021, 2022 | | | |
| | Causal recipe(s) | ML1\|ML2 | | | |
| Colombia | Configurations | [0,0,0,0,0] | [0,0,1,0,0] | | |
| | Years | 2010,2011 | 2012,2013,2014, 2015,2016,2017, 2018,2019,2020, 2021,2022 | | |
| | Causal recipe(s) | ML2 | ML1\|ML2 | | |
| Korea | Configuration | [0,0,0,0,0] | [1,0,0,0,0] | [1,1,0,0,0] | |
| | Years | 2010,2011,2012, 2013,2014,2015, 2016,2017,2018, 2019,2020 | 2021 | 2022 | |
| | Causal recipe(s) | ML2 | — | MH1 | |

Selected typical countries are analyzed via a country-specific membership lens. For each country-year in the dataset, the most relevant configurations are noted, along with the causal recipes linking them to High-HCE or Low-HCE.

surge in 2020, HCE had not returned to 2019 levels by 2022, remaining relatively high. Denmark followed a similar trajectory to Austria but with a lag; its High-HCE recipe was MH1 from 2012, switching to both MH1 and MH3 from 2018 to 2022. It started to decline from 2021 to 2022 but still remained at a relatively high level.

Belgium's High-HCE causal recipe shifted from MH1 to MH2 between 2013 and 2021, characterized by high healthcare costs. During the pandemic, Belgium experienced the smallest increase in HCE compared to other countries and quickly saw a decline in HCE starting in 2021. Ireland's High-HCE recipe was MH2 from 2014 to 2022, with a 3% drop in HCE in 2014. The significant rise in income from 2014, due to GDP growth, was a key factor, illustrating that an increase in GDP can drive a decrease in HCE..

In the United States, the High-HCE recipe was MH2 from 2010 to 2022, with a notable increase in the elderly population from 2022. This suggests that the High-HCE in the United States is mainly due to a large proportion of unhealthy people and high healthcare costs, with the elderly population playing a less significant role.

Low-HCE was primarily associated with less developed economies. ML2 (low economic development, low proportion of the elderly population, low healthcare provision, and low healthcare costs) was the most relevant Low-HCE recipe in the observed years. For Chile, the Low-HCE recipes were ML1 and ML2. Compared to Colombia, Chile had a larger proportion of unhealthy people, leading to higher HCE from 2012. In Colombia, HCE declined before switching to both ML1 and ML2 from 2012, influenced by an increase in the unhealthy population, leading to a rising trend in HCE. Compared to developed countries, the COVID-19 pandemic had a lesser impact on undeveloped countries, which experienced a more rapid recovery in HCE.

Korea stands out as an interesting case. Its causal recipe was ML2 from 2010 to 2020, with high income emerging in 2021 and a high proportion of the elderly population in 2022, transitioning to MH1 in 2022. This shift moved Korea from a Low-HCE to a High-HCE country, showing a continuous rising trend. In contrast to the United States, a large proportion of the elderly population appears to be the principal driver of High-HCE, partially explaining the varying importance of aging population influences on HCE found in previous studies.

## Conclusions

This study provides a comprehensive country-level understanding of HCE and its influencing factors through a novel approach using panel data and fuzzy set Qualitative Comparative Analysis (fsQCA). Spanning 13 years (2010–2022), this longitudinal analysis offers robust insights into the trends and evolution of HCE and its determinants over time. The study makes several key contributions:

Firstly, with respect to fsQCA, this research extends previous findings from single-year analyses to a multiyear panel data context. This approach not only identifies causal recipes but also examines time effects for selected countries. Utilizing 481 country-year observations from 37 countries, the panel data analysis demonstrates the potential of fsQCA for "longitudinal set-theoretic research." The technical BECONS and WICONS consistency measures provide nuanced insights at both the year and country levels, supporting the study's findings.

Secondly, the study offers novel insights into the factors influencing HCE. Despite the relative stability of the applicable recipes, there are notable fluctuations in the recipes associated with specific countries, especially those with varying demographic structures and High-HCE. These fluctuations vary in duration and the changes in countries' associated recipes. Additionally, we find that factors such as aging populations and income growth can have

different impacts on HCE in various contexts, supporting the differing findings of previous studies.

Thirdly, the study highlights the heterogeneous experiences of developing and developed economies, both as groups and as individual entities, following the 2019 pandemic crisis. It suggests that policy differences between countries may explain these variations and warrant further investigation. The analysis also identifies countries most relevant for further study, based on the evolution of income, unhealthy populations, aging population, healthcare provision and healthcare price.

However, the study has limitations. It focuses solely on OECD countries, which may not be representative of other nations. Additionally, it does not explore the reasons behind changes in recipes and configurations in specific countries or how these changes might relate to evolving policies, indicating a need for further research.

Despite these limitations, the study demonstrates that fsQCA can effectively evaluate the interaction of country-level factors with HCE over an extended period. This offers valuable insights for policymakers regarding the effectiveness of different factors and HCE trends. The novelty of this longitudinal analysis, as opposed to previous single-year studies, allows for a dynamic evaluation of HCE factors. The policy and managerial implications are significant, particularly for governments in economies with large aging populations and high HCE. The findings suggest a need for constant review of conditions and supporting policies to manage HCE fluctuations likely caused by pandemics.

## Declaration of generative AI and AI-assisted technologies in the writing process

During the preparation of this work the authors used chatGPT in order to improve language and readability. After using this tool/service, the authors reviewed and edited the content as needed and take full responsibility for the content of the publication.

## Data availability statement

All data are available from the OECD database (URL: https://data-explorer.oecd.org/?lc=en) and WHO database (URL: https://www.who.int/data/collections). 1. The URL for the indicator HCE as a share of GDP: https://data-explorer.oecd.org/vis?lc=en&fs[0]=Topic%2C1%7CHealth%23HEA%23%7CHealth%20expenditure%20and%20financing%23HEA_EXP%23&pg=0&fc=Topic&bp=true&snb=5&vw=tb&df[ds]=dsDisseminateFinalDMZ&df[id]=DSD_SHA%40DF_SHA&df[ag]=OECD.ELS.HD&df[vs]=1.0&dq=AUT%2BBEL%2BCAN%2BCHL%2BCOL%2BCRI%2BCZE%2BDNK%2BEST%2BFIN%2BFRA%2BDEU%2BGRC%2BHUN%2BISL%2BIRL%2BISR%2BITA%2BJPN%2BKOR%2BLVA%2BLTU%2BLUX%2BMEX%2BNLD%2BNZL%2BNOR%2BPOL%2BPRT%2BSVK%2BSVN%2BESP%2BSWE%2BCHE%2BTUR%2BGBR%2BUSA%2BAUS.A.EXP_HEALTH.PT_B1GQ._T.._T.._T...&pd=2010%2C2022&to[TIME_PERIOD]=false. 2. The URL for the indicator GDP per capital in USD: https://data-explorer.oecd.org/vis?lc=en&fs[0]=Topic%2C0%7CEconomy%23ECO%23&pg=0&fc=Topic&bp=true&snb=279&vw=tb&df[ds]=dsDisseminateFinalDMZ&df[id]=DSD_NAAG%40DF_NAAG_I&df[ag]=OECD.SDD.NAD&df[vs]=1.0&dq=A.AUS%2BAUT%2BBEL%2BCAN%2BCHL%2BCOL%2BCRI%2BCZE%2BDNK%2BEST%2BFIN%2BFRA%2BDEU%2BGRC%2BHUN%2BISL%2BIRL%2BISR%2BITA%2BJPN%2BKOR%2BLVA%2BLTU%2BLUX%2BMEX%2BNLD%2BNZL%2BNOR%2BPOL%2BPRT%2BSVK%2BSVN%2BESP%2BSWE%2BCHE%2BTUR%2BGBR%2BUSA.B1GQ_POP.USD_PPP_PS.&pd=2010%2C2022&to[TIME_PERIOD]=false. 3. The URL for the indicator Population at age 65 or

over: https://data-explorer.oecd.org/vis?lc=en&fs[0]=Topic%2C0%7CSociety%23SOC%23&fs[1]=Topic%2C1%7CSociety%23SOC%23%7CDemography%23SOC_DEM%23&pg=0&fc=Topic&snb=2&vw=tb&df[ds]=dsDisseminateFinalDMZ&df[id]=DSD_POPULATION%40DF_POP_HIST&df[ag]=OECD.ELS.SAE&df[vs]=1.0&dq=GRC%2BHUN%2BFIN%2BFRA%2BDEU%2BEST%2BDNK%2BCZE%2BCRI%2BCOL%2BCHL%2BISL%2BIRL%2BISR%2BITA%2BKOR%2BJPN%2BLVA%2BLTU%2BLUX%2BMEX%2BNLD%2BNZL%2BNOR%2BPOL%2BPRT%2BSVK%2BSVN%2BESP%2BSWE%2BCHE%2BTUR%2BGBR%2BCAN%2BUSA%2BBEL%2BAUT%2BAUS..PT_POP._T.Y_GE65.&pd=2010%2C2022&to[TIME_PERIOD]=false. 4. The URL for the indicator Population of obese: https://www.who.int/data/gho/data/indicators/indicator-details/GHO/prevalence-of-obesity-among-adults-bmi--30-(age-standardized-estimate)-(-). 5. The URL for the indicator Active doctors per 1000 inhabitants: https://data-explorer.oecd.org/vis?lc=en&tm=active%20physician&pg=0&snb=9&vw=tb&df[ds]=dsDisseminateFinalDMZ&df[id]=DSD_HEALTH_EMP_REAC%40DF_PHYS&df[ag]=OECD.ELS.HD&df[vs]=1.0&dq=AUT%2BBEL%2BCAN%2BCHL%2BCOL%2BCZE%2BDNK%2BEST%2BFIN%2BFRA%2BDEU%2BGRC%2BHUN%2BISL%2BIRL%2BISR%2BITA%2BJPN%2BKOR%2BLVA%2BLTU%2BLUX%2BMEX%2BNLD%2BNZL%2BNOR%2BPOL%2BPRT%2BSVK%2BSVN%2BESP%2BSWE%2BCHE%2BTUR%2BGBR%2BUSA%2BAUS..10P3HB....._T%2BP.&pd=2010%2C2022&to[TIME_PERIOD]=false. 6. The URL for the indicator Price level in healthcare sector: https://data-explorer.oecd.org/vis?lc=en&fs[0]=Topic%2C1%7CEconomy%23ECO%23%7CPrices%23ECO_PRI%23&fs[1]=Frequency%20of%20observation%2C0%7CAnnual%23A%23&fs[2]=Measure%2C0%7CPrice%20level%23PL%23&fs[3]=Analytical%20categories%2C2%7CGDP%23A0%23%7CActual%20individual%20consumption%23A01%23%7CHealth%23A0106%23&fs[4]=Base%20reference%20area%2C0%7COECD%23OECD%23&pg=0&fc=Base%20reference%20area&snb=2&vw=tb&df[ds]=dsDisseminateFinalDMZ&df[id]=DSD_PPP%40DF_PPP_CPL&df[ag]=OECD.SDD.TPS&df[vs]=1.0&dq=AUT%2BBEL%2BCAN%2BCHL%2BCOL%2BCRI%2BCZE%2BDNK%2BEST%2BFIN%2BFRA%2BDEU%2BGRC%2BHUN%2BISL%2BIRL%2BISR%2BITA%2BJPN%2BKOR%2BLVA%2BLTU%2BLUX%2BMEX%2BNLD%2BNZL%2BNOR%2BPOL%2BPRT%2BSVK%2BSVN%2BESP%2BSWE%2BCHE%2BTUR%2BGBR%2BUSA%2BAUS.A.PL.A0106..OECD&lom=LASTNPERIODS&lo=10&to[TIME_PERIOD]=false. Origin data is also available at https://github.com/nieshr/hce-research-data/blob/main/raw.csv.

## Author contributions

**Data curation:** Shanren Nie.

**Supervision:** Dong Liu, Sheng Chen.

**Writing – original draft:** Shanren Nie.

**Writing – review & editing:** Dong Liu.

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
