## [Decision Letter · Decision Letter 0]

18 Mar 2025

PONE-D-24-40627Complex interactions in Healthcare Expenditure through the years: A panel data analysis using fsQCA in OECD countriesPLOS ONE

Dear Dr. chen,

Thank you for submitting your manuscript to PLOS ONE. After careful consideration, we feel that it has merit but does not fully meet PLOS ONE’s publication criteria as it currently stands. Therefore, we invite you to submit a revised version of the manuscript that addresses the points raised during the review process.

We look forward to receiving your revised manuscript.

Kind regards,

Saurav Guha, Ph.D.

Academic Editor

PLOS ONE

Journal Requirements:

2. Please note that your Data Availability Statement is currently missing [the repository name and/or the DOI/accession number of each dataset OR a direct link to access each database]. If your manuscript is accepted for publication, you will be asked to provide these details on a very short timeline. We therefore suggest that you provide this information now, though we will not hold up the peer review process if you are unable.

Additional Editor Comments:

This manuscript presents a longitudinal study examining interactions in healthcare expenditure. One reviewer asked for a detailed mathematical derivations on linear interpolation of missing data points. Without some notations and derivations, it is hard to read. The authors can include this in the appendix. Another reviewer highlighted the limited description of the methodology. It is really tough to link the method and the results discussed in the manuscript. Moreover, author should include some motivating examples justifying the use of fuzzy-set QCA. Authors have discussed some results about necessary and suffiicent conditions. These conditions should be described in the methodology. The discussion section is very much straightforward and requires further enhancement for new insights.

Reviewers' comments:

Reviewer's Responses to Questions

**Comments to the Author**

1. Is the manuscript technically sound, and do the data support the conclusions?

Reviewer #1: Yes

Reviewer #2: Yes

2. Has the statistical analysis been performed appropriately and rigorously? 

Reviewer #1: Yes

Reviewer #2: Yes

3. Have the authors made all data underlying the findings in their manuscript fully available?

Reviewer #1: Yes

Reviewer #2: Yes

4. Is the manuscript presented in an intelligible fashion and written in standard English?

Reviewer #1: Yes

Reviewer #2: No

5. Review Comments to the Author

Reviewer #1: In Sub-section 2.2, the authors mentioned that "Missing data points were estimated using linear interpolation". It is recommended that this should be presented more in details (mathematical derivations) in Appendix.

Reviewer #2: The authors are requested to address the following points for clarification:

1. The authors should justify the placement of the sufficiency analysis process in the Results section, as it may be more appropriately included in the Methods section. Specifically, they should explain why the process of sufficiency analysis is presented in the Results section rather than in the Methods section. Additionally, clarification is needed on why certain study findings have been included in the Discussion section instead of the Results section.

2. The manuscript should include line numbers to facilitate the review process.

6. PLOS authors have the option to publish the peer review history of their article (what does this mean?). If published, this will include your full peer review and any attached files.

Reviewer #1: No

Reviewer #2: No

---

## [Author Response · Author response to Decision Letter 1]

1 Apr 2025

Dear editor,

Thank you for your time and effort in reviewing my manuscript. I have carefully addressed each point raised by the reviewers and have made the necessary revisions accordingly. Details are specified in Response to reviewers.

Best regards.

---

## [Decision Letter · Decision Letter 1]

28 Apr 2025

Complex interactions in Healthcare Expenditure through the years: A panel data analysis using fsQCA in OECD countries

PONE-D-24-40627R1

Dear Dr. chen,

We’re pleased to inform you that your manuscript has been judged scientifically suitable for publication and will be formally accepted for publication once it meets all outstanding technical requirements.

Kind regards,

Saurav Guha, Ph.D.

Academic Editor

PLOS ONE

Additional Editor Comments (optional):

Reviewers' comments:

Reviewer's Responses to Questions

**Comments to the Author**

1. If the authors have adequately addressed your comments raised in a previous round of review and you feel that this manuscript is now acceptable for publication, you may indicate that here to bypass the “Comments to the Author” section, enter your conflict of interest statement in the “Confidential to Editor” section, and submit your "Accept" recommendation.

Reviewer #1: All comments have been addressed

Reviewer #2: All comments have been addressed

2. Is the manuscript technically sound, and do the data support the conclusions?

Reviewer #1: Yes

Reviewer #2: Yes

3. Has the statistical analysis been performed appropriately and rigorously? 

Reviewer #1: Yes

Reviewer #2: Yes

4. Have the authors made all data underlying the findings in their manuscript fully available?

Reviewer #1: Yes

Reviewer #2: Yes

5. Is the manuscript presented in an intelligible fashion and written in standard English?

Reviewer #1: Yes

Reviewer #2: Yes

6. Review Comments to the Author

Reviewer #1: (No Response)

Reviewer #2: (No Response)

7. PLOS authors have the option to publish the peer review history of their article (what does this mean?). If published, this will include your full peer review and any attached files.

Reviewer #1: No

Reviewer #2: **Yes: **Basant Adhikari

---

## [Editor Report · Acceptance letter]

PONE-D-24-40627R1

PLOS ONE

Dear Dr. chen,

I'm pleased to inform you that your manuscript has been deemed suitable for publication in PLOS ONE. Congratulations! Your manuscript is now being handed over to our production team.

Kind regards,

on behalf of

Dr. Saurav Guha

Academic Editor

PLOS ONE